Knowledge of acute stroke management and the predictors among Malaysian healthcare professionals

http://orcid.org/0000-0002-0206-2470 Albart Stephenie Ann 1
Yusof Khan Abdul Hanif Khan 2 ahanifkhan@upm.edu.my
Abdul Rashid Aneesa 3
Wan Zaidi Wan Asyraf 4
Bidin Mohammad Zulkarnain 5
Looi Irene 1 6
Hoo Fan Kee 2
1 Clinical Research Centre, Ministry of Health Malaysia, Hospital Seberang Jaya , Seberang Jaya, Penang , Malaysia
2 Department of Neurology, Faculty of Medicine and Health Sciences, Universiti Putra Malaysia , Serdang, Selangor , Malaysia
3 Department of Family Medicine, Faculty of Medicine and Health Sciences, Universiti Putra Malaysia , Serdang, Selangor , Malaysia
4 Department of Medicine, Hospital Canselor Tuanku Muhriz, Universiti Kebangsaan Malaysia , Kuala Lumpur , Malaysia
5 Department of Medicine, Faculty of Medicine and Health Sciences, Universiti Putra Malaysia , Serdang, Selangor , Malaysia
6 Department of Medicine, Ministry of Health Malaysia, Hospital Seberang Jaya , Seberang Jaya, Penang , Malaysia
Levine David
Electronic publication date: 2022 Apr 20
Publication date: 2022
Volume: 10
Electronic Location ID: e13310
Received 2021 Nov 8; Accepted 2022 Mar 30
Copyright: © 2022 Albart et al.
Copyright year: 2022
Copyright holder: Albart et al.
License: This is an open access article distributed under the terms of the Creative Commons Attribution License, which permits unrestricted use, distribution, reproduction and adaptation in any medium and for any purpose provided that it is properly attributed. For attribution, the original author(s), title, publication source (PeerJ) and either DOI or URL of the article must be cited.
License URL: https://creativecommons.org/licenses/by/4.0/

Keywords: Stroke, Acute stroke management, Knowledge, Healthcare professional, Predictors, Questionnaire

Funding: Boehringer Ingelheim This study received the Educational Grant from Boehringer Ingelheim under the ANGELS Initiative. The funders had no role in study design, data collection and analysis, decision to publish, or preparation of the manuscript.

==============================
Background

Despite rapid advances in acute ischaemic stroke (AIS) management, many healthcare professionals (HCPs) might not be aware of the latest recommended management of AIS patients. Therefore, we aimed to determine the level and factors associated with AIS management knowledge among Malaysian HCPs.

Methods

This cross-sectional online questionnaire study was conducted nationwide among 627 HCPs in Malaysia using the Acute Stroke Management Questionnaire (ASMaQ). Multiple logistic regression was used to predict the relationship between the independent variables (age, gender, years of service, profession, work setting, work sector, seeing stroke patients in daily practice, and working with specialists) and the outcome variable (good vs poor knowledge).

Results

Approximately 76% (95% CI [73–79%]) of HCPs had good overall knowledge of stroke. The highest proportion of HCPs with good knowledge was noted for General Stroke Knowledge (GSK) [88.5% (95% CI [86–91%])], followed by Advanced Stroke Management (ASM) [61.2% (95% CI [57–65%])] and Hyperacute Stroke Management (HSM) [58.1% (95% CI [54–62%])]. The odds of having poor knowledge of stroke were significantly higher among non-doctor HCPs [adjusted OR = 3.46 (95% CI [1.49–8.03]), P = 0.004]; among those not seeing stroke patients in daily practice [adjusted OR = 2.67 (95% CI [1.73–4.10]), P < 0.001]; and among those working without specialists [adjusted OR = 2.41 (95% CI [1.38–4.18]), P = 0.002].

Conclusions

Stroke education should be prioritised for HCPs with limited experience and guidance. All HCPs need to be up-to-date on the latest AIS management and be able to make a prompt referral to an appropriate facility. Therefore, more stroke patients will benefit from advanced stroke care.

Introduction

Seventy-five percent of all stroke-related deaths globally occurred in low- and middle-income countries, and Asian countries had the highest lifetime stroke risk from 1990 to 2016 (The GBD 2016 Lifetime Risk of Stroke Collaborators, 2018). In Malaysia, stroke is the second leading cause of morbidity and the third leading cause of mortality (Institute for Health Metrics and Evaluation, 2021).

Despite rapid advances in acute ischaemic stroke (AIS) management, many stroke patients do not receive treatment at the appropriate time (Hashim et al., 2013). Healthcare professional (HCP) related factors might lead to suboptimal stroke care (Hashim et al., 2013). Although a new clinical practice guideline (CPG) on AIS management was available in Malaysia in 2021, there had been a 10-year gap since the last CPG was published, which may have led to a lack of awareness of the latest management (Academy of Medicine of Malaysia, 2021).

There have been very few studies published in the past 5 years on AIS management knowledge among HCPs, and none from Malaysia. The target population of the studies has mainly been general practitioners, paramedics, and emergency nurses. Previous studies had a wide variation in HCPs’ responses to the stroke knowledge questions, and most of them recommended stroke education to improve their knowledge (Lin et al., 2017; Shahzad et al., 2018; Yeganeh et al., 2019; Chang et al., 2020; Blek & Szarpak, 2021; Kusuma et al., 2021).

Years of clinical experience (Lin et al., 2017; Chang et al., 2020; Blek & Szarpak, 2021; Kusuma et al., 2021), profession (Rababah, Al-Hammouri & AlNsour, 2021), working in stroke-related specialities (Mellon et al., 2015), the number of beds (Yeganeh et al., 2019), and the number of stroke cases (Blek & Szarpak, 2021) were known to be associated factors of stroke management knowledge. However, previous studies did not explore factors such as the work sector (government vs private), work settings (hospital vs primary care clinic), and the presence of guidance from specialists.

Therefore, the researchers aimed to determine the level and factors associated with AIS management knowledge among Malaysian HCPs.

Materials and Methods

Study setting

This study involved both public and private healthcare facilities in Malaysia. As of December 2019, there are 154 public hospitals, 208 private hospitals, 1,114 government health clinics, and 7,988 private medical clinics in Malaysia (Portal Rasmi Kementerian Kesihatan Malaysia, 2022).

The healthcare system in Malaysia comprises primary, secondary, and tertiary care. Primary care focuses on general health services such as health promotion, disease prevention, health maintenance, counselling, patient education, diagnosis, and treatment of acute and chronic illnesses as an outpatient. It is also the basis for referral to secondary and tertiary care hospitals if highly specialised equipment and expertise are required. Primary care providers, especially in the public sector, usually consist of family medicine specialists, medical doctors, and other allied health professionals, whereas in the private sector, they consist of doctors and clinic assistants.

There are two types of secondary care services provided in Malaysia, namely basic and full secondary care. The basic secondary care services are general medicine, general surgery, obstetrics and gynaecology, and paediatrics. They are available at district hospitals and overseen by resident medical officers and visiting specialists. Full secondary care services include additional services such as orthopaedics, anesthesiology, psychiatry, dermatology, medical rehabilitation, pathology, imaging, dental, otorhinolaryngology, opthalmology, and geriatrics. They are available in some district and general hospitals and are overseen by resident specialists and medical officers.

Tertiary care is highly specialised and provides services such as cardiology, cardiothoracic surgery, paediatric surgery, neurology, neurosurgery, respiratory medicine, urology, nephrology, plastic surgery & burns, maxilofacial, haematology, radiotherapy, oncology, and endocrinology. They are mainly available at private hospitals, university hospitals, and some general hospitals (Official Portal of Economic Planning Unit, 2020).

Study design

A cross-sectional online questionnaire study was conducted nationwide among HCPs in Malaysia from February until July 2021. HCPs are defined as medical practitioners and allied healthcare professionals from various medical disciplines. A convenience sampling method followed by the snowball sampling method was used to recruit participants. The online questionnaire link was emailed to all HCPs registered for the Stroke e-learning Module on the Docquity platform. The interested participants were informed to fill in the online questionnaire before accessing the module to get the baseline knowledge scores.

Ethical approval

This study was approved by the Medical Research and Ethics Committee (MREC) of the Ministry of Health Malaysia (NMRR-20-2706-57567). Informed consent was obtained from all participants.

Data measurement

The online questionnaire link consisted of two sections. Section 1 (Demographic information) collected data on independent variables such as age, gender, years of service, profession, work setting (primary care vs hospital), work sector (private vs government), seeing stroke patients in daily practice, and working with specialists. Section 2 (Acute Stroke Management Questionnaire (ASMaQ)) measures the outcome variables. The ASMaQ was used to measure the knowledge of AIS management among HCPs. It consisted of 29 items and had three domains: General Stroke Knowledge (GSK), Hyperacute Stroke Management (HSM), and Advanced Stroke Management (ASM). Cronbach’s alpha for the overall ASMaQ was 0.82 (Sim et al., 2021). The items were scored using a five-point Likert scale ranging from 1 to 5.

Outcome variable

The outcome variable, knowledge scores, was categorised into good and poor. The cutoff point was determined based on expert opinions from stroke neurologists, internal medicine physicians, and family medicine specialists actively involved in stroke management. Marks were converted to positive scores for those negative answers as the desired response, with a higher score indicating better knowledge. The scores of 1 to 3 were categorised into poor knowledge, and 4 to 5 were categorized into good knowledge. Therefore, a score of 3.5 was considered a cutoff point for good and poor knowledge, and the score range for the level of knowledge of each domain was classified accordingly under Table S1.

Statistical methods

We used IBM SPSS Version 20 for statistical analysis. We estimated the proportion of HCPs with good and poor knowledge of stroke with a 95% confidence interval (CI). We performed a binary logistic regression analysis to predict the relationship between independent variables and knowledge outcomes. First, we ran simple logistic regression on each independent variable, followed by multiple logistic regression to control for confounding effects. Variables are selected based on automatic Forward and Backward Likelihood Ratio (LR) methods. The preliminary model was checked for possible two-way interactions and multicollinearity problems. The goodness-of-fit assessment was done using the Hosmer–Lemeshow test, based on the classification table and area under the Receiving Operating Characteristics curve (AUC) methods. All the hypotheses involved were two-sided tests. Independent variables with a P value of less than 0.05 in the final model were considered statistically significant predictors of poor knowledge. We also analysed the 95% CI of odds ratios.

Study size

We set α = 0.05 and ρ = 2 (standard deviation that would include all possible values for two levels categorical variable (good vs poor knowledge)) (Adam, 2020). With the estimated HCP population size of 200,000, the minimum returned sample size required at 95% CI and a margin of error of 0.05 was 384. The total number of participants who answered the questionnaire during the study period was 627; which was the final sample size.

Results

A total of 627 participants who answered the online questionnaire were included in the study. The median (inter-quartile range) for age and years of service were 32 (seven) and seven (eight), respectively. Both variables were skewed to the right; therefore, they were grouped into three-level categories using quantiles. Table 1 describes the baseline characteristics of participants. The participants were predominantly female (63%), medical officers (65.6%), working in the hospital (66.8%), in the government sector (82.6%), seeing stroke patients in daily practice (77.8%), and working with specialists (80.5%).

Table 1 Baseline characteristics.

Variables	 	Total n (%)	
Age group (years)	≤30
31-35
≥36	217 (34.6)
209 (33.3)
201 (32.1)	
Gender	Female
Male	395 (63.0)
232 (37.0)	
Service group (years)	<5
5-10
>10	210 (33.5)
241 (38.4)
176 (28.1)	
Professions	Others§
General practitioner (GP)
House officer (HO)
Medical officer (MO)
Specialist	40 (6.4)
57 (9.1)
20 (3.2)
411 (65.6)
 99 (15.8)	
Medical officers based on department	Medical based dept.¥
Surgical based dept.
Primary care	263 (64.0)
27 (6.6)
121 (29.4)	
Specialists based on department	Medical based dept.¥
Surgical based dept.
Primary care	68 (68.7)
15 (15.2)
16 (16.2)	
Work setting	Primary care clinic
Hospital	208 (33.2)
419 (66.8)	
Types of primary care clinics	Government health clinic
Private clinic	138 (66.3)
70 (33.7)	
Types of hospitals	Private hosp.
General hosp.
District hosp.
University hosp.	39 (9.3)
202 (48.2)
134 (32.0)
44 (10.5)	
Work sector	Private
Government	109 (17.4)
518 (82.6)	
Seeing stroke patients in daily practice	No
Yes	139 (22.2)
488 (77.8)	
Workplace with specialists	No
Yes	122 (32.1)
505 (80.5)	
Notes:

§ Others: medical assistants, nurses, pharmacists and allied health professionals

¥ Medical based department was mostly involved the General Medicine and Emergency department.

Healthcare professionals’ level of stroke knowledge

Table 2 shows the proportion of participants with good and poor knowledge and the total mean scores. About 76% (95% CI [73–79%]) of HCPs had good overall knowledge of stroke. The highest proportion of HCP with good knowledge was noted for General Stroke Knowledge (GSK) [88.5% (95% CI [86–91%])], followed by Advanced Stroke Management (ASM) [61.2% (95% CI [57–65%])] and Hyperacute Stroke Management (HSM) [58.1% (95% CI [54–62%])]. The overall knowledge total mean score was 107.2 (SD = 9.29) and the total mean scores for GSK, HSM, and ASM were 39.18 (3.4), 32.43 (3.67), and 36.32 (5.35), respectively.

Table 2 The proportion of participants with good and poor knowledge and the total mean scores.

Knowledge	Good knowledge	Poor knowledge	Total mean scores	
n	% (95% CI)	n	% (95% CI)	Mean (SD)	95% CI	
General Stroke Knowledge (GSK)	555	88.5 [86–91]	72	11.5 [9–14]	39.18 (3.40)	[38.9–39.4]	
Hyperacute Stroke Management (HSM)	364	58.1 [54–62]	263	41.9 [38–46]	32.43 (3.67)	[32.1–32.7]	
Advanced Stroke Management (ASM)	384	61.2 [57–65]	243	38.8 [35–43]	36.32 (5.35)	[35.9–36.7]	
Overall	476	75.9 [73–79]	151	24.1 [21–27]	107.20 (9.29)	[106–108]	
Note:

CI, Confidence Interval; SD, Standard deviation.

For GSK among HCPs, the highest proportion of poor responses (undesired and neutral responses) was noted for item GSK-10 (Acute stroke management education should be conducted regularly for healthcare professionals) (99.9%) and followed by GSK-8 (A full neurological examination must be performed immediately in patients presenting acutely with symptoms suggestive of stroke) (96.3%); see Fig. 1; Table S2.

Figure 1 Participants’ responses (%) to general stroke knowledge.

An asterisk (*) sign denotes a negative answer as desired response.

For HSM, poor knowledge was noted for item HSM-8 (All acute stroke patients must have a 12 leads ECG before thrombolysis) (93.1%), HSM-7 (Coagulation profile must be screened before thrombolysis) (87.7%), and HSM-1 (Stroke is a medical emergency only within 4.5 h of stroke onset) (70%); see Fig. 2; Table S2.

Figure 2 Participants’ responses (%) to hyperacute stroke management.

An asterisk (*) sign denotes a negative answer as desired response.

For ASM, the poor response was for item ASM-6 (My hospital is equipped with mechanical thrombectomy service) (85.1%), ASM-10 (Wake up strokes are not eligible for thrombolysis nor mechanical thrombectomy) (73.2%), ASM-8 (Mechanical thrombectomy can be performed after thrombolysis therapy) (62%) and ASM-3 (How would you rate your knowledge of acute stroke management?) (50.2%); see Fig. 3; Table S2.

Figure 3 Participants’ responses (%) to advanced stroke management.

An asterisk (*) sign denotes a negative answer as desired response.

The factors associated with knowledge of stroke among healthcare professionals

Overall knowledge of stroke

Multiple logistic regression showed that the professions, not seeing stroke patients in daily practice, and working without specialists were significant predictors of poor overall stroke knowledge among HCPs. Compared to a specialist, non-doctor HCPs had 3.46 times the odds of poor knowledge (P = 0.004). However, the odds of poor knowledge were not statistically significant among general practitioners, house officers, medical officers, and specialists. Those not seeing stroke patients in daily practice had 2.67 times the odds of poor knowledge than those seeing stroke patients (P < 0.001). Those not working with specialists in their workplace had 2.41 times the odds of poor knowledge than those without specialists (P = 0.002) (Table 3).

Table 3 Predictors of poor knowledge of stroke among healthcare professionals.

Variables	Knowledge	Simple Logistic Regression	Multiple Logistic Regression	
Good n (%)	Poor n (%)	Crude OR (95% CI)	X2 stat. (df)a	P-valuea	Adjusted OR
(95% CI)	X2 stat. (df)a	P-valuea	
Age group (years)
≤30
31-35
≥36	
162 (74.7)
162 (77.5)
152 (75.6)	
55 (25.3)
47 (22.5)
49 (24.4)	
1.05 (0.68,1.64)
0.90 (0.57,1.42)
1	0.492 (2)
0.052 (1)b
0.204 (1)b	0.782
0.819b
0.651b	–	–	–	
Gender
Female
Male	
295 (74.7)
181 (78.0)	
100 (25.3)
51 (22.0)	
1.20 (0.82,1.77)
1	
0.896 (1)	
0.344	–	–	–	
Service group (years)
<5
5-10
>10	
153 (72.9)
189 (78.4)
134 (76.1)	
57 (27.1)
52 (21.6)
42 (23.9)	
1.19 (0.75,1.89)
0.88 (0.55,1.40)
1	1.9 (2)
0.539 (1)b
0.304 (1)b	0.387
0.463b
0.581b	–	–	–	
Professions
Others§
GP
HO
MO
Specialist	
22 (55.0)
39 (68.4)
17 (85.0)
316 (76.9)
82 (82.8)	
18 (45.0)
18 (31.6)
3 (15.0)
95 (23.1)
17 (17.2)	
3.95 (1.75,8.90)
2.23 (1.04,4.78)
0.85 (0.22,3.23)
1.45 (0.82,2.57)
1	13.974 (4)
10.956 (1)b
4.208 (1)b
0.056 (1)b
1.630 (1)b	0.007
0.001b
0.04b
0.813b
0.202b	
3.46 (1.49,8.03)
0.76 (0.29,1.97)
0.96 (0.25,3.74)
1.40 (0.77,2.54)
1	12.371 (4)
8.313 (1)b
0.323 (1)b
0.002 (1)b
1.23 (1)b	0.015
0.004b
0.570b
0.962b
0.267b	
Work setting
Primary care
Hospital	
140 (67.3)
336 (80.2)	
68 (32.7)
83 (19.8)	
1.97 (1.35,2.87)
1	
12.238 (1)	
<0.001	–	–	–	
Work sector
Private
Government	
78 (71.6)
398 (76.8)	
31 (28.4)
120 (23.2)	
1.32 (0.83,2.10)
1	
1.33 (1)	
0.249	–	–	–	
Seeing stroke patients in daily practice
No
Yes	

83 (59.7)
393(80.5)	

56 (40.3)
95 (19.5)	

2.79 (1.86,4.19)
1	

23.737 (1)	

<0.001	

2.67 (1.73,4.10)
1	

19.49 (1)	

<0.001	
Workplace with specialists
No
Yes	
76 (62.3)
400 (79.2)	
46 (37.7)
105 (20.8)	
2.31 (1.51,3.53)
1	
14.275 (1)	
<0.001	
2.41 (1.38,4.18)
1	
9.364 (1)	
0.002	
Notes:

Abbreviations: GP= General Practitioner; HO= House Officer; MO= Medical Officer; OR=Odds Ratio; CI= Confidence Interval; df= Degree of freedom

a Likelihood Ratio test

b Wald test

§ Others: medical assistants, nurses, pharmacists and allied health professionals

There was no significant interaction found between the predictors. There was no multicollinearity problem identified (variance-inflation-factor less than 10), which indicated a statistically stable model. The Hosmer–Lemeshow test was insignificant (P = 0.932), which showed that the dataset fits well with the logistic model. The classification table showed that 77% of cases are predicted correctly, whether they have poor knowledge or not, with 97.3% specificity and 13.2% sensitivity. The AUC was 0.66 (95% CI [0.61–0.71]), and it was an acceptable fit to discriminate against poor vs good knowledge (Fig. S1). All methods of goodness-of-fit showed a good model fit. Cook’s influential statistic showed no influential outlier present (all the data points were below the cutoff point of 1.0). The Nagelkerke R square showed 10.3% of the variation in the outcome variable was explained by the model.

General stroke knowledge

The profession was a significant predictor of poor GSK knowledge (P < 0.001). Other non-doctor HCPs had 6.67 times the odds of poor knowledge compared to specialists (Table 4).

Table 4 Predictors of poor knowledge of General Stroke Knowledge (GSK).

Variables	Knowledge	Simple Logistic Regression	Multiple Logistic Regression	
Good n (%)	Poor n (%)	Crude OR (95% CI)	X2 stat. (df)a	P-valuea	Adjusted OR (95% CI)	X2 stat. (df)a	P-valuea	
Age group (years)
≤30
31-35
≥36	
188 (86.6)
191 (91.4)
176 (87.6)	
29 (13.4)
18 (8.6)
25 (12.4)	
1.09 (0.61,1.93)
0.66 (0.35,1.26)
1	2.734 (2)
0.080 (1)b
1.581 (1)b	0.255
0.778b
0.209b	–	–	–	
Gender
Female
Male	
350 (88.6)
205 (88.4)	
45 (11.4)
27 (11.6)	
0.93 (0.59,1.62)
1	
0.009 (1)	
0.926	–	–	–	
Service group (years)
<5
5-10
>10	
187 (89.0)
214 (88.8)
154 (87.5)	
23 (11.0)
27 (11.2)
22 (12.5)	
0.86 (0.46,1.60)
0.88 (0.49,1.61)
1	0.252 (2)
0.222 (1)b
0.165 (1)b	0.881
0.637b
0.685b	–	–	–	
Profession
Others§
GP
HO
MO
Specialist	
24 (60.0)
47 (82.5)
17 (85.0)
377 (91.7)
90 (90.1)	
16 (40.0)
10 (17.5)
3 (15.0)
34 (8.3)
9 (9.1)	
6.67 (2.62,16.94)
2.13 (0.81,5.60)
1.77 (0.43,7.20)
0.90 (0.42,1.95)
1	28.465 (4)
15.898 (1)b
2.341 (1)b
0.627 (1)b
0.069 (1)b	<0.001
<0.001b
0.126b
0.428b
0.793b	
6.67 (2.62,16.94)
2.13 (0.81,5.60)
1.77 (0.43,7.20)
0.90 (0.42,1.95)
1	28.465 (4)
15.898 (1)b
2.341 (1)b
0.627 (1)b
0.069 (1)b	<0.001
<0.001b
0.126b
0.428b
0.793b	
Work setting
Primary care
Hospital	
177 (85.1)
378 (90.2)	
31 (14.9)
41 (9.8)	
1.62 (0.98,2.66)
1	
3.458 (1)	
0.063	–	–	–	
Work sector
Private
Government	
94 (86.2)
461 (89.0)	
15 (13.8)
57 (11.0)	
1.29 (0.70,2.38)
1	
0.646 (1)	
0.422	–	–	–	
Seeing stroke patient in daily practice
No
Yes	

116 (83.5)
439 (90.0)	

23 (16.5)
49 (10.0)	

1.78 (1.04,3.04)
1	

4.175 (1)	

0.041	–	–	–	
Workplace with specialist
No
Yes	
104 (85.2)
451 (89.3)	
18 (14.8)
54 (10.7)	
1.45 (0.81,2.57)
1	
1.508 (1)	
0.207	–	–	–	
Notes:

Abbreviations: GP= General Practitioner; HO= House Officer; MO= Medical Officer; OR=Odds Ratio; CI= Confidence Interval; df= Degree of freedom

a Likelihood Ratio test

b Wald test

§ Others: medical assistants, nurses, pharmacists and allied health professionals

Hyperacute stroke management

For HSM, those without or unsure of the availability of thrombolysis services in their workplace had 9.52 times the odds of poor knowledge than those working in a workplace with thrombolysis services (P < 0.001). Other factors were not statistically significant (Table 5).

Table 5 Predictors of poor knowledge of Hyperacute Stroke Management (HSM).

Variables	Knowledge	Simple Logistic Regression	Multiple Logistic Regression	
Good n (%)	Poor n (%)	Crude OR (95% CI)	X2 stat. (df)a	P-valuea	Adjusted OR (95% CI)	X2 stat. (df)a	P-valuea	
Age group (years)
≤30
31-35
≥36	
122 (56.2)
128 (61.2)
114 (56.7)	
95 (43.8)
81 (38.8)
87 (43.3)	
1.02 (0.69,1.50)
0.83 (0.56,1.23)
1	1.326 (2)
0.01 (1)b
0.868 (1)b	0.515
0.919b
0.352b	–	–	–	
Gender
Female
Male	
223 (56.5)
141 (60.8)	
91 (39.2)
172 (43.5)	
1.20 (0.86,1.66)
1	
1.123 (1)	
0.289	–	–	–	
Service group (years)
<5
5-10
>10	
112 (53.3)
153 (63.5)
99 (56.2)	
98 (46.7)
88 (36.5)
77 (43.8)	
1.13 (0.75,1.68)
0.74 (0.50,1.10)
1	5.101(2)
0.329 (1)
2.222 (1)	0.078
0.566
0.136	–	–	–	
Profession
Others§
GP
HO
MO
Specialist	
26 (65.0)
28 (49.1)
9 (45.0)
235 (57.2)
66 (66.7)	
14 (35.0)
29 (50.9)
11 (55.0)
176 (42.8)
33 (33.3)	
1.08 (0.40,2.33)
1.06 (1.06,4.03)
0.92 (0.92,6.48)
0.94 (0.94,2.38)
1	7.246 (4)
0.035 (1)
4.59 (1)
3.23 (1)
2.95 (1)	0.123
0.851b
0.032b
0.072b
0.086b	–	–	–	
Work setting
Primary care
Hospital	
99 (47.6)
265 (63.2)	
109 (52.4)
154 (36.8)	
1.90 (1.35,2.65)
1	 
13.9 (1)	
<0.001	–	–	–	
Work sector
Private
Government	
52 (47.7)
312 (60.2)	
57 (52.3)
206 (39.8)	
1.66 (1.10,2.51)
1	
5.734 (1)	
0.017	–	–	–	
Seeing stroke patient in daily practice
No
Yes	

71 (58.1)
293 (60.0)	

68 (48.9)
195 (40.0)	

1.44 (0.99,2.10)
1	

3.539 (1)	

0.06	–	–	–	
Workplace with specialist
No
Yes	
48 (39.3)
316 (62.6)	
74 (60.7)
189 (37.4)	
2.58 (1.72,3.87)
1	
21.527 (1)	
<0.001	–	–	–	
Workplace with thrombolysis services
No/Unsure
Yes	

91 (31.3)
273 (81.2)	

200 (68.7)
63 (18.8)	

9.52 (6.58,13.78)
1	

167.0 (1)	

<0.001	

9.52 (6.58,13.78)
1	

167.0 (1)	

<0.001	
Notes:

Abbreviations: GP= General Practitioner; HO= House Officer; MO= Medical Officer; OR=Odds Ratio; CI= Confidence Interval; df= Degree of freedom

a Likelihood Ratio test

b Wald test

§ Others: medical assistants, nurses, pharmacists and allied health professionals

Advanced stroke management

For ASM, significant factors for poor knowledge were gender (P < 0.001), service group (P = 0.015), seeing stroke patients in daily practice (P < 0.001), workplace with specialists (P < 0.001), and availability of thrombectomy services in the workplace (P < 0.001). Female HCPs had 1.98 times the odds of poor knowledge than males. Those in service for less than 5 years had 1.73 times the odds of poor knowledge than those over 11 years (P = 0.02). However, the odds of poor knowledge were insignificant among those with 5–10 years and more than 10 years of service (P = 0.959). Those not seeing stroke patients in daily practice had 2.55 times the odds of poor knowledge than those seeing stroke patients. Those not working with specialists had 2.68 times the odds of poor knowledge than those without specialists. Those not having or unaware of thrombectomy services in their workplace had 24.78 times the odds of poor knowledge than those workplaces without the services (Table 6).

Table 6 Predictors of poor knowledge of Advanced Stroke Management (ASM).

Variables	Knowledge	Simple logistic regression	Multiple logistic regression	
Good n (%)	Poor n (%)	Crude OR (95% CI)	X2 stat. (df)a	P-valuea	Adjusted OR (95% CI)	X2 stat. (df)a	P-valuea	
Age group (years)
≤30
31-35
≥36	
121 (55.8)
139 (66.5)
124 (61.7)	
96 (44.2)
70 (33.5)
77 (38.3)	
1.28 (0.86,1.89)
0.81 (0.54,1.22)
1	5.211 (2)
1.511 (1)
1.032 (1)	0.074
0.219b
0.310b	–	–	–	
Gender
Female
Male	
224 (56.7)
160 (69.0)	
172 (43.3)
72 (31.0)	
1.70 (1.21,2.39)
1	
9.382 (1)	
0.002	
1.98 (1.35,2.90)
1	
12.72 (1)	
<0.001	
Service group (years)
<5
5-10
>10	
115 (54.8)
158 (65.6)
111 (63.1)	
95 (45.2)
83 (34.4)
65 (36.9)	
1.41 (0.94,2.12)
0.90 (0.60,1.34)
1	5.823 (2)
2.715 (1)
0.276 (1)	0.054
0.099b
0.599b	
1.73 (1.09,2.74)
0.99 (0.63,1.55)
1	8.39 (2)
5.42b (1)
0.003b (1)	0.015
0.02b
0.959b	
Profession
Others§
GP
HO
MO
Specialist	
22 (55.0)
25 (43.9)
14 (70.0)
248 (60.3)
75 (75.8)	
18 (45.0)
32 (56.1)
6 (30.0)
163 (39.7)
24 (24.2)	
2.56 (1.18,5.55)
4.00 (1.99,8.03)
1.34 (0.46,3.87)
2.05 (1.25,3.39)
1	17.859 (4)
5.649 (1)
15.222 (1)
0.291 (1)
7.95 (1)	0.001
0.017b
<0.001b
0.589b
0.005b	–	–	–	
Work setting
Primary care
Hospital	
102 (49.0)
282 (67.3)	
106 (51.0)
137 (32.7)	
2.14 (1.52,3.01)
1	
19.328 (1)	
<0.001	–	–	–	
Work sector
Private
Government	
57 (52.3)
327 (63.1)	
52 (47.7)
191 (36.9)	
1.56 (1.03,2.37)
1	
4.377 (1)	
0.036	–	–	–	
Seeing stroke patient in daily practice
No
Yes	

63 (45.3)
321 (65.8)	

76 (54.7)
167 (34.2)	

2.32 (1.58,3.40)
1	

18.676 (1)	

<0.001	

2.55 (1.65,3.94)
1	

18.01 (1)	

<0.001	
Workplace with specialist
No
Yes	
50 (41.0)
334 (66.1)	
72 (59.0)
171 (33.9)	
2.81 (1.88,4.22)
1	
25.574 (1)	
<0.001	
2.68 (1.70,4.20)
1	
18.83 (1)	
<0.001	
Workplace with thrombectomy services
No/Unsure
Yes	

294 (55.1)
90 (96.8)	

240 (44.9)
3 (3.2)	

24.49 (7.66,78.34)
1	

75.91 (1)	

<0.001	

24.78 (7.62,80.66)
1	

69.15 (1)	

<0.001	
Notes:

Abbreviations: GP= General Practitioner; HO= House Officer; MO= Medical Officer; OR=Odds Ratio; CI= Confidence Interval; df= Degree of freedom

a Likelihood Ratio test

b Wald test

§ Others: medical assistants, nurses, pharmacists and allied health professionals

Discussion

Comprehensively, this is the first study to use ASMaQ as a tool to assess HCPs’ knowledge of AIS management. Unfortunately, most tool items were not similar to other studies to compare our findings, especially on the HSM and ASM responses. For GSK, HSM, and ASM discussion, the items with the negative answer as the desired response were italicized in the brackets.

General stroke knowledge

The majority of poor responses to the GSK domain were noted on GSK-10 and GSK-8. The negative answer was the desired response for GSK-10 (Acute stroke management education should be conducted regularly for healthcare professionals). Item GSK-10 was created to assess the adequacy of stroke knowledge among respondents indirectly, and low scores indicated a lack of stroke knowledge, thus requiring regular education (Sim et al., 2021). Our study reported that almost all participants agreed with this item.

For GSK-8 (A full neurological examination must be performed immediately in patients presenting acutely with symptoms suggestive of stroke), 96% of HCPs wrongly perceived that a full neurological examination was required. NIHSS is sufficient for the initial assessment of acute stroke patients (Academy of Medicine of Malaysia, 2021; Powers et al., 2019). Therefore, poor knowledge regarding this assessment might cause a delay in managing stroke patients.

Responses to GSK-9 (High blood pressure must be lowered to normal values in acute stroke) showed that 25.5% of HCPs were still not aware or perceived that high blood pressure (BP) should be lowered to normal values in acute stroke. Elevated BP after a stroke should be carefully lowered to at or below 185/110 mmHg before the initiation of acute therapy; otherwise, it should not be treated in acute stroke patients below 220/120 mmHg (Academy of Medicine of Malaysia, 2021). Previously, it was reported that 80% of paramedics were not managing hypertension according to guidelines (Blek & Szarpak, 2021). Another study reported that 62.6% of stroke patients were over-treated with anti-hypertensive (Haidar et al., 2021). In a study done among GPs, 27% wanted a drastic BP reduction that could be harmful to acute stroke patients, and 14% of GPs were unsure of the ideal BP target (Shahzad et al., 2018). Even though the majority of our participants are aware of proper blood pressure management in AIS and it is better than in previous studies, it should be further improved as it is one of the absolute contraindications for the initiation of acute reperfusion therapy.

Almost 90% of our participants had good knowledge of items GSK-1 to GSK-7 and were well aware of the signs and symptoms of stroke. Our finding was also compared with previous studies, whereby 85% of nurses were able to recognise stroke symptoms (Yeganeh et al., 2019), while among GPs, knowledge of the signs and symptoms of a stroke ranged from 79% to 99% (Chang et al., 2020). The previous study among GPs and nurses with similar items to GSK-3, GSK-4, and GSK-5 reported a much lower percentage of knowledge, which was 15.4%, 66.8%, and 13.6%, respectively (Yang et al., 2015).

The profession was associated with the GSK level in our study, whereby non-doctor HCPs had the poorest knowledge, mainly as they were not on the front line of stroke management.

Hyperacute stroke management

The responses to items HSM-8, HSM-7, and HSM-1 were poor. About 93% answered incorrectly or were unsure about item HSM-8 (All acute stroke patients must have a 12 leads ECG before thrombolysis). About 88% answered incorrectly or were unsure about HSM-7 (Coagulation profile must be screened before thrombolysis). Our finding showed that most HCPs had poor knowledge of the investigations indicated for the initiation of thrombolysis. Twelve-lead ECG and coagulation profiles are required in all stroke patients, but they are not compulsory and should not delay the initiation of thrombolysis in AIS patients (Academy of Medicine of Malaysia, 2021; Powers et al., 2019).

For HSM-1 (Stroke is a medical emergency only within 4.5 h of stroke onset), 70% of our participants were unaware of thrombolysis above 4.5 h. Intravenous thrombolysis can also be considered for acute stroke onset more than 4.5 h up to 9 h or in wake-up stroke or stroke of uncertain onset assisted by CT/MR perfusion (significant penumbra mismatch) or MRI (DWI-FLAIR mismatch) (Academy of Medicine of Malaysia, 2021). Our finding is similar to that reported previously, especially regarding the treatment of extended hours of stroke; most participants were unaware of stroke treatment beyond 4.5 h (Blek & Szarpak, 2021). Thus, potentially treatable stroke patients could be missed due to poor knowledge.

For HSM-6 (My hospital is equipped with thrombolysis treatment), 32.9% do not have thrombolysis treatment in their workplace, and 13.6% were unaware of it. Our study identified that those HCPs had significantly poor knowledge of HSM as they might not have experience managing hyperacute stroke. There were about 362 hospitals (154 public and 208 private) in Malaysia (Portal Rasmi Kementerian Kesihatan Malaysia, 2022), but only 48 hospitals (24 public and 24 private) offered thrombolysis service (Chia et al., 2020). Thus, the actual percentage of HCPs with poor knowledge of HSM in Malaysia might be higher than that reported in this study.

Advanced stroke management

The majority of poor responses within the ASM domain were for ASM-6, ASM-10, and ASM-8. About 66% of HCPs stated that thrombectomy services were unavailable in their workplace. Our findings from HSM-6 and ASM-6 suggest that Malaysia needs more facilities with thrombolysis and thrombectomy services. Thus, HCPs can increase their knowledge and experience in managing acute strokes, which will benefit many stroke patients.

For item ASM-10 (Wake up strokes are not eligible for thrombolysis nor mechanical thrombectomy), half of the HCPs were unsure of the eligibility of AIS therapy in wake-up strokes, and 27% had wrongly perceived that it was not eligible for treatment. Again, this brings up the issue of HCPs not being aware of stroke treatment in the extended window period.

For item ASM-8 (Mechanical thrombectomy can be performed after thrombolysis therapy), 62% of HCPs were unaware of combination therapy of thrombolysis and thrombectomy. Thrombectomy can be performed after thrombolysis in eligible AIS patients who arrive within 4.5 h of stroke onset (Academy of Medicine of Malaysia, 2021). Previously, it was reported that 86% of physicians were unaware of the possibility of thrombectomy with thrombolysis, which was higher compared to our study (Gatto et al., 2017).

For ASM, predictors of poor knowledge were short duration of services, not seeing stroke patients in daily practice, working without specialists, and unavailability or being unaware of thrombectomy services in the workplace. This could lead to poor knowledge and would be a barrier to proper stroke management. A systematic review showed that organisational factors such as lack of stroke specialists and health professionals’ lack of awareness, knowledge, and skills in acute management were the common barriers to stroke care practise (Baatiema et al., 2017). Gender was also found to be a significant factor affecting ASM knowledge in our study. This might be due to the unequal distribution of male and female participants in this study. Besides, this association might be due to other confounding factors that were not explored in this study.

Overall knowledge of stroke

Generally, most of our HCPs had good knowledge of stroke, but it could be further improved, especially on HSM and ASM aspects. The majority of them were not up-to-date on the current indications of thrombolysis and thrombectomy, particularly regarding the extended hours.

This study found that HCPs who saw stroke patients in daily practice, especially doctors and those working with specialists, had better overall knowledge of stroke, showing that experience and proper guidance are the determinant factors of stroke knowledge. Similarly, previous studies reported that being a physician (Rababah, Al-Hammouri & AlNsour, 2021), working in a stroke-related speciality (Mellon et al., 2015), and seeing more stroke cases (Blek & Szarpak, 2021), are significantly associated with higher stroke knowledge. Furthermore, training, skill, and expertise in acute stroke care were significantly associated with higher thrombolysis rates (Paul et al., 2015). Thus, having a better knowledge of stroke management will lead to a better outcome for stroke patients.

Training HCPs on stroke management is one of the strategies to improve the quality of care for stroke patients (Rababah, Al-Hammouri & AlNsour, 2021; Baatiema et al., 2020). As a part of a nationwide initiative to improve stroke care in Malaysia, an online stroke e-learning module was launched to raise awareness of AIS management among HCPs (Docquity.com, 2021). The module was developed based on the latest Malaysian clinical practice guideline in collaboration with the ANGELs Initiative. This module may help our HCPs to improve their knowledge of stroke management; however, further research is needed to assess the module’s effectiveness.

Limitations

This study has several limitations. Firstly, response bias could influence the estimation of the proportion of HCPs with poor knowledge. Those who responded are more likely to be involved in stroke management and might have better knowledge than non-responders. Secondly, our study does not have an equal representation of HCPs due to the non-probability sampling methods used in this study. Therefore, future research should consider probability sampling for less bias and more generalizable findings. Thirdly, ASMaQ has certain limitations. Most of the items in the questionnaire are general. It did not include the thrombolytic medication dosages and contraindications for AIS treatment. However, highly specific questions were omitted during the validation of this questionnaire to keep the instrument relevant throughout time (Sim et al., 2021). Finally, even though most of our HCPs were found to have good knowledge of AIS management, this might not reflect their actual practice. Therefore, further research is needed to evaluate their practice in AIS management.

Conclusions

About one-fourth of HCPs had poor knowledge of AIS management, particularly in hyperacute and advanced stroke management. Mostly, they were not aware of the AIS treatment during extended hours. Stroke knowledge was significantly poor among HCPs who do not see stroke patients on a daily basis, do not work with specialists, and work in facilities that do not offer thrombolysis and thrombectomy services. Our findings reinforce the need for stroke education on the current recommended AIS management, especially for HCPs with limited experience and guidance in managing stroke patients. All HCPs need to be aware of the latest AIS management and make a prompt referral to an appropriate facility. Therefore, more stroke patients will benefit from advanced stroke care. Getting appropriate healthcare fast is crucial, and knowing where to find those resources quickly is of utmost importance.

Supplemental Information

Supplemental Information 1 Classification of level of knowledge.

Click here for additional data file.

Supplemental Information 2 Participants’ responses and scores for the ASMaQ.

Click here for additional data file.

Supplemental Information 3 Area under the ROC curve - shows the model’s ability to discriminate between good and poor knowledge for overall knowledge of stroke.

Click here for additional data file.

Supplemental Information 4 Raw Data.

The participants’ responses for the online questionnaire.

Click here for additional data file.

We thank the Director-General of Health Malaysia for permission to publish these research findings. We would also like to thank Docquity Malaysia for providing a platform for the Stroke e-learning Module.

Additional Information and Declarations

Competing Interests

Author Contributions

Human Ethics

Data Availability

The authors declare that they have no competing interests.

Stephenie Ann Albart conceived and designed the experiments, performed the experiments, analyzed the data, prepared figures and/or tables, authored or reviewed drafts of the paper, and approved the final draft.

Abdul Hanif Khan Yusof Khan conceived and designed the experiments, performed the experiments, analyzed the data, prepared figures and/or tables, authored or reviewed drafts of the paper, and approved the final draft.

Aneesa Abdul Rashid conceived and designed the experiments, performed the experiments, analyzed the data, prepared figures and/or tables, authored or reviewed drafts of the paper, and approved the final draft.

Wan Asyraf Wan Zaidi conceived and designed the experiments, performed the experiments, analyzed the data, prepared figures and/or tables, authored or reviewed drafts of the paper, and approved the final draft.

Mohammad Zulkarnain Bidin conceived and designed the experiments, performed the experiments, analyzed the data, prepared figures and/or tables, authored or reviewed drafts of the paper, and approved the final draft.

Irene Looi conceived and designed the experiments, performed the experiments, analyzed the data, prepared figures and/or tables, authored or reviewed drafts of the paper, and approved the final draft.

Fan Kee Hoo conceived and designed the experiments, performed the experiments, analyzed the data, prepared figures and/or tables, authored or reviewed drafts of the paper, and approved the final draft.

The following information was supplied relating to ethical approvals (i.e., approving body and any reference numbers):

The Medical Research and Ethics Committee (MREC) Ministry of Health Malaysia (MOH) has provided ethical approval for this study.

The following information was supplied regarding data availability:

The raw data are available in the Supplemental File.

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
