# Peer review of "Knowledge of acute stroke management and the predictors among Malaysian healthcare professionals"

_PeerJ, doi:10.7717/peerj.13310_

## Round 0.1 · original submission · Minor Revisions

Dear authors, there are minor revisions that need to be addressed before acceptance, please see the reviews.

Reviewer 1 ·

Basic reporting

No comment

Experimental design

Can you explain the setting of healthcare system in Malaysia under study settings/design. The difference between primary and hospital care, what’s their functions and personnel?
How many district hospitals, university hospitals were involved? How many hospitals that have stroke units were involved? What kind of specialists/physicians were involved, from which departments were residents involved from? How many private vs the government hospitals were involved?

Validity of the findings

No comment

Additional comments

This is a good and relevant article since stroke is one of the leading causes of disability and mortality in low- and middle-income countries. Despite this, there are some clarifications that are needed.
1. Different healthcare professionals have different roles under stroke management. For example, doctors are responsible for neurological examinations, interpretation of CT/MRI images, and initiating thrombolysis after excluding contraindications, while nurses nurse patients, therefore they do not have the same roles. Therefore, it is difficult to compare them.

2. Looking at the general knowledge of stroke management questionnaire (GSK) and some other few questions are the only parts where one can compare different healthcare professionals since it deals with among others stroke symptoms which are crucial for everyone. Otherwise, other parts of the questionnaire are specific for specific healthcare cadres.

3. Under discussion, some parts of the questionnaire (e.g., stroke symptoms, thrombolytic therapy window, etc) can be compared to some previous studies. Compare with some previous studies that have assessed similar parts, e.g.,
Yang J, Zhang J, Ou S, Wang N, Wang J. Knowledge of Community General Practitioners and Nurses on Pre-Hospital Stroke Preven tion and Treatment in Chongqing, China. PLOS ONE 2015; 10: e0138476.

4. Under limitations: Not all questions on stroke management were included in the questionnaire e.g., thrombolytic medication dosages, contraindications, etc
5. Results - Factors associated with knowledge: Most of the overall knowledge of stroke findings are results of the subsections (GSK, ASM) such that they could be incorporated under it.

Reviewer 2 ·

Basic reporting

69: Rose over what timeframe?
73-74: Change to “Healthcare professionals’ related…” No hyphen needed
76: Change to “possibly” or “likely contributing”
86: Comma after [7]
107: Comma before and
112: For readability, add “The” at the start of the sentence.
113: Change to a :
115: Wording with transition between ASM and Cronbach’s alpha is confusing. The survey as a whole has the alpha level described or just that section? Perhaps two sentences.
115: Replace “They” with “The items” or something descriptive vs the pronoun.
230: Reword sentence about gender differences
124: Although scores are plural, the noun is variable so should be “was categorized” as it is currently written. Or adjust the sentence to flip the wording of Knowledge scores were categorized…, which were utilized as the outcome variable.
262-263: Wording is awkward. The patient isn’t being lowered, but their elevated BP is. Therefore, reword to “Elevated BP after a stroke should carefully be lowered at …”
264: Needing a verb. “…paramedics were not managing”
279: on the front line
283: answering incorrectly

Experimental design

Experimental Design
Strengths: Research question well defined as was the gap in the literature to guide this study.
87: Good job identifying a gap in the literature to form the foundation of the study
116: Perhaps introduce this chapter with the two descriptions of the online survey, and then as section 2 is being described, include the information that is currently 112-115.
354: According to the start of the methods section, the online course mailing list was used to recruit subjects. Is it known if the course had been completed? Or is there a measurement tool (pre/post) for the effectiveness of the training? If all participants have taken the training, it would be anticipated that the knowledge is higher than other medical professionals. Likewise, in the limitation section, if the participants were part of a stroke education mailing list, this would likely bias the results as discussed. This is also an opportunity for replication by using a different means to sample a larger representation of the population that may have even less experience with people with strokes.

Validity of the findings

Validity of Findings
Strengths:
It is important to see that even with practitioners that are seeing patients with strokes at times and have registered for education, still are lacking in areas of advanced stroke management. This has implications for a broader sample that isn’t recruited from a stroke mailing list. A follow-up or replication study could investigate practitioners with wider backgrounds. The findings also reinforce the need for continued education, especially for individuals who do not regularly treat people with stroke. Getting appropriate healthcare fast is key and knowing where to find those resources quickly is of utmost importance.
248: Why would a negative response be preferred with GSK-10? Is it linked to that previous education was happening but changes in performance were not being reflected in the workplace? For practitioners who are not regularly treating patients with stroke, it seems regular education would be beneficial. This may involve including discussion from the initial articles from creating the tool in describing this question. Line 256 does a nice job explaining why the negative answer is correct for GSK-8, this would be helpful for the GSK-10 segment. Line 350 states that training is a strategy to improve quality of care, which seems to contradict this test item.
197 (and throughout): Replace insignificant with not statistically significant or not statistically significant findings because being not significant can often be important.
268: 14% of GPs?
287: Good discussion again on why the item is incorrect with the evidence to back-up the correct answer to the question.
295: Opportunity for further education for readers – perhaps include how long the treatment can be provided beyond the 4.5 hours or what instances allow for increased treatment.
331: Unequal number of women/men in the study, also what was the breakdown of women vs men as far as roles? I appreciate that it is noted that there might be other factors, but those may need to be explored in greater detail or at least identified.

Additional comments

91 (and throughout): Remove “we”. Change to something like “Therefore, the researchers' goal for the study was to determine…” or “The researchers aimed”
100: Similar to the previous comment: “A convenience sample followed by snowballing was used…”
370: While it is unsurprising that the individuals with the lowest scores see the fewest patients with stroke, the question is what does this mean long-term or what does this mean for Malaysia? If a facility rarely sees individuals with stroke and does not have access to advanced care, is the goal for those practitioners to be able to identify symptoms and make appropriate referrals? Or is the goal to increase facilities with advanced care available? Education is still the key component in either scenario.

---

## Round 0.2 · Minor Revisions

Please make the additional revisions that reviewer one has suggested

Reviewer 1 ·

Basic reporting

English grammar: the article needs to be corrected so that its of high quality. If you have any programme like grammarly, it will assist you.

Other sections are satisfactory, but more comments on additional comments category below

Experimental design

Satisfactory

Validity of the findings

Satisfactory

Additional comments

Review comments
Thank you for addressing and clarifying my queries/ questions, the manuscript is now taking shape and getting better. Despite that, there are some parts that need further clarification.

More review comments
Data measurements
1.Section 2: ASMaQ. This is the first time it is mentioned, therefore should be written in full with the abbreviation in brackets.
2. Outcome variables
Grammar: Outcome scores were categorized into … not as

Results
3. Level of knowledge on stroke among healthcare professionals
The highest proportion of good knowledge was noted on GSK (88.5%), followed by ASM (61.2%) and HSM (58.1%) (Table 2).
Was this significant, if so where was the significance between GSK and ASM, or GSK and HSM?
For overall knowledge score, you mentioned the mean score with confidence interval. Would be best to mention the other mean scores with confidence intervals too.

4. Paragraph 2 (208-212): Rephrasing the sentence will make it easier to read and understand e.g. For GSK among HCPs, the highest proportion of poor responses (undesired and neutral response) was noted on item GSK-10 [Acute stroke management education should be conducted regularly for healthcare professionals] (99.9%)…..
The same applies to paragraph 3 (214-217): For HSM, poor knowledge was noted on item HSM-8 [All acute stroke patients must have a 12 leads ECG before thrombolysis] (93.1%),….
Also paragraph 3 (219-223), paragraph 4 (229-231), .
Mention also confidence intervals of the mean scores unless if your text/wording exceed the requirements

5. GSK
Paragraph 1 (252-253): Rephrasing/rearranging of words to make the point visible and easily understandable. Also, include p-value.

6. ASM
Paragraph 1 (264-266): repetition of ASM (at the start of the sentence and at the end). Rearrange the sentence e.g., For ASM, significant factors for poor knowledge were ….
Also, include p-value.

7. It is best to report results using past tense, not mixing past and present tenses

Discussion
8. Some of the texts in brackets are in italics while some are not, is there any reason? Lines 333, 343, 359, 364 etc.

9. Thrombectomy should be performed after thrombolysis in large vessel occlusion in AIS patients who arrive within 6 hours……
Can thrombectomy not be given without thrombolysis?

10. Line 400: should be module not modules

11. Line 371-373: It needs rephrasing to make the point clearer and easy understandable e.g., For ASM, predictors of poor knowledge were short duration of services, not seeing stroke patients in daily practices, working without specialists, and unavailability or being unaware of thrombectomy services in the workplace

Other
12. Missing subsections before references: Information on funding, competing interests, availability of data and materials, and author contributions

Thank you in advance

---

## Round 0.3 · accepted · Accept

The last version of the document seems to make all of the corrections that reviewers wanted, thank you for making the changes.